# Quality Control of Drinking Water in the City of Ilave, Region of Puno, Peru

**DOI:** 10.3390/ijerph191710779

**Published:** 2022-08-30

**Authors:** Pompeyo Ferro, Luis Jhordan Rossel-Bernedo, Ana Lucia Ferró-Gonzáles, Ivone Vaz-Moreira

**Affiliations:** 1Universidad Nacional Intercultural Fabiola Salazar Leguía de Bagua, Jr. Ancash 520, Bagua 01721, Amazonas, Peru; 2Universidad Privada San Carlos, Sede Ilave, Jr. Ilo 343, Ilave 21501, Puno, Peru; 3Laboratorios Bioproyect SAC, Jr. Tacna 344, Puno 21000, Puno, Peru; 4Universidad Nacional de Juliaca, Av. Nueva Zelandia 631, Juliaca 21101, Puno, Peru; 5Universidade Católica Portuguesa, CBQF—Centro de Biotecnologia e Química Fina—Laboratório Associado, Escola Superior de Biotecnologia, Rua Diogo Botelho 1327, 4169-005 Porto, Portugal

**Keywords:** drinking water system, sanitary control, water quality, disinfection

## Abstract

The region of Puno, in Peru, is described as a region with some health conditions that may be associated with the water quality, such as a high index of anemia or cases of acute diarrhea in children. This study aimed at monitoring the drinking water quality of the city of Ilave, in Peru, and determining possible correlations between physical-chemical and microbiological parameters, and the water distribution conditions, such as the period of water availability. Physical-chemical parameters (turbidity, residual chlorine, temperature, conductivity, and pH), microbiological parameters (presence of coliforms), and heavy metals (Zn, Mn, Ni, Fe, and Cu) were determined. All the parameters quantified were within the maximum permissible limits according to Peruvian regulations, except for residual chlorine, which was, for all the treated water samples, below the recommended value of 0.5 mg/L. Coliforms that should be absent from drinking water were detected in all the household samples. These results demonstrate the need for the inclusion of additional steps of re-chlorination along the distribution system to guarantee the maintenance of residual levels of chlorine that assure the microbiological quality of water. The quality of the drinking water was not observed to correlate with the period of water availability.

## 1. Introduction

Diseases associated with water are a leading cause of morbidity and mortality in developing countries [1,2]. Therefore, an adequate supply of drinking water with good quality to drink and wash food is critical to reducing the transmission of diseases [3]. Diseases related to inadequate water sanitation and hygiene are also a huge economic burden for developing countries [4]. Therefore, knowing the quality of water for human consumption is of utmost importance in the development of a country [5].

To avoid waterborne diseases, governments promote the disinfection of drinking water through sanitary regulations. The use of chlorine as a disinfectant is among the most commonly used treatments [6,7]. One of the big advantages of chlorine as a disinfectant is the residual effect that allows a longer effect and monitoring, which has been adopted and recommended by the World Health Organization [2,7]. Chlorine is added to drinking water to reduce or eliminate microorganisms causing waterborne diseases, and should be applied at different points of the drinking water distribution system [8,9], guaranteeing the destruction of any agent that may be introduced after the treatment [10]. The monitoring of the residual chlorine concentration in situ is one of the parameters that is measured frequently, to assess the microbial safety of the water [11].

In 2018, just 52% of the Peruvian households had access to safe drinking water, with there being observed inequalities in the access according to the size of the city and the economic condition of the house owners [12]. In Peru, it is still very common that the water service is intermittent with interruptions in the distribution. The stoppage of the water pump is a way to reduce the expense of electrical power [13]. However, it may result in the degradation of the quality of the water supplied to the users [14]. In peri-urban communities, fecal contamination of drinking water is still very common in Peru [15,16,17,18,19].

Anemia indexes are extremely alarming in the Region of Puno, where, of every ten children, seven have anemia. In addition to this, acute diarrheal diseases cause infant mortality [20,21,22,23]. One of the factors that was correlated with these circumstances was the water quality since, according to the WHO, half of the anemia cases are due to the consumption of water without any type of disinfection. 

This study aimed at monitoring the drinking water quality of the city of Ilave, in the Region of Puno, Peru, and determining possible correlations between physical-chemical and microbiological parameters and the water distribution conditions, such as the period of water availability.

## 2. Materials and Methods 

### 2.1. Delimitation of the Study and Sampling

Ilave, the capital of the province of El Collao, is located at a distance of 50 km from the city of Puno, at 3850 m above sea level in the central Andes plateau (Collao plateau). It has a total surface area of 874.57 km^2^. Due to its geographical location, the climate throughout the year is typical of the highlands, frigid, dry, and temperate, due to the presence of Lake Titicaca, having slight variations according to each season (average temperature: 8–15 °C; average annual precipitation: 725 mm).

The drinking water distribution system considered in this study is managed by the Sanitation Services and Administrative Management Unit (UGASS, Unidades de Gestión Administrativa de Servicios de Saneamiento) of the city of Ilave. The drinking water system is conventional, with water being collected from the Ilave river, distributed by gravity and treated by flocculation, decantation, filtration, and disinfection (chlorination through a drip dispenser or constant flow with pumping). The drinking water distribution network has an extension of 48.7 km, covering about 60% of the population (9358 inhabitants, with more than 6083 household water connections). The continuity of service, in the sampled houses, varies between 4 to 8 h per day. 

The water sampling was carried out during October 2019, on three consecutive weeks (11, 18, and 25 October), and three measurements of each parameter were evaluated. One of the samples was collected at the river Ilave (PUNTO 1) and the others (PUNTO 2–PUNTO 10) collected from household taps. The household water samples were collected from the kitchen tap, to have a water sample representative of the water that is consumed. The sampling was done following the standard protocols for sampling, preservation, transport, storage, and analysis of water for human consumption, according to the Peruvian Ministry of Health [24]. The sampling points are detailed in Table 1 and Figure 1.

### 2.2. Evaluated Parameters 

All the samples were analyzed for the mandatory control parameters according to Peruvian regulations [25]: physical-chemical (turbidity, residual chlorine, temperature, conductivity, and pH), microbiological (total coliforms, fecal coliforms, and *E. coli*), and heavy metals (Zn, Mn, Ni, Fe, and Cu). The legal limits accepted in water used for human consumption are indicated in Table 2. The physical-chemical parameters were measured at the collection point, while the microbiological and heavy metals analyses were performed at the laboratory, at maximum 4 h after collection.

HI9828 Multiparameter (HANNA Instruments Ltd., Bedfordshire, UK) equipment was used for the determination of pH, conductivity, and temperature. TN-100 portable Turbidimeter (EUTECH Instruments, Singapore) was used to measure turbidity, and for residual chlorine, POCKET II CLORIMETER (HACH Company, Loveland, USA) was used, using DPD (diethyl-p-phenyldiamine) tablets as a reagent for the measurement. For the heavy metal analysis, the HI 83200 multiparameter bench photometer (HANNA Instruments Ltd., Bedfordshire, UK) was used, using the reagents and protocols recommended by the manufacturer. The maximum limits of quantification were of 3.0 mg/L for Zn, 0.3 mg/L for Mn, 1.0 mg/L for Ni, 0.4 mg/L for Fe, and 5 mg/L for Cu.

For the microbiological analysis, the portable MEL Total Coliform and *E. coli* Laboratory equipment (HACH Company, Loveland, CO, USA) was used. All the containers were previously sterilized and the ones to be used to collect treated water (PUNTO 2–PUNTO 10) were prepared with 10 mg of sodium thiosulfate per 100 mL of sample, to neutralize the chlorine. Volumes of 100 mL of water sample were used. The m-ColiBlue 24 broth (M00PMCB24, (HACH Company, Loveland, USA) was used to detect and quantify total coliforms and *E. coli* after 24 h of incubation at 35 °C and 44.5 °C, respectively. The m-FC, without rosolic acid, broth (MHA00FCR2, Millipore, MA, USA) was used for the detection and quantification of fecal coliforms after 24 h of incubation at 44.5 °C.

### 2.3. Statistical Analyses

The results of the different parameters studied were compared among samples based on the parametric test T-student or the non-parametric Kruskal–Wallis test, depending on if the results followed a normal distribution or not, and on the ANOVA with the post-hoc Tukey test. The existence of a linear correlation between physical-chemical parameters was determined using the Pearson correlation coefficient. All the statistical analyses were performed with the SPSS software package, version 28.0 (IBM SPSS software, Chicago, IL, USA). In addition, a principal components analysis (PCA) was carried out to understand the samples distribution and correlations between physical-chemical parameters.

## 3. Results

### 3.1. Physical-Chemical Parameters

Nine household taps and a river, used as the source for drinking water production, were sampled. All the drinking water samples fulfilled the quality criteria for human consumption in terms of the physical-chemical parameters determined [25]. However, some variances were observed between the three sampling days, with the most notorious being a significant increase (*p* < 0.05) in the temperature from the first and second (A and B) to the third sampling date (C) (Figure 2). As expected, a negative correlation (r = −0.758, *p* < 0.01) was observed between the temperature increase and the pH values (Figure 3b), which tendentially decreased in sampling C. The same negative correlation was observed in the principal components analysis (Figure 3a).

The concentration of residual chlorine in treated water was low (0–0.62 mg/L) in all the sampling points with a random variation along samplings. In Punto 1 (river water), no chlorine was detected, as expected. Turbidity ranged from 0.31–2.73 NTU, being positively correlated (r = 0.375, *p* < 0.05) with the pH (Figure 3). Conductivity did not vary between samplings, with the exception of Punto 1, 4, and 10, being commonly determined to be 600–650 µhmo/cm.

Regarding heavy metals, Zn was not detected and Ni was just detected in two samples, samples A from Punto 2 and 3 (Figure 2). Fe, Mn, and Cu were detected in the drinking water, below the maximum permissible limits (Figure 2, Table 1) in the range of 0.034–0.201 mg/L, 0–0.17 mg/L, and 0–1.21 mg/L, respectively. Two of the points with the highest concentration of Cu (Punto 5 and 7) were closely located (Figure 1). However, the geographical location does not seem to be the determinant, since Punto 6 also has a high concentration of Cu and is in a different area of the city (Figure 1).

Punto 2–4 have a water availability period of 4 h, while Punto 5–10 have water for 8 h a day. However, no significant correlation (*p* > 0.05) was observed between the period of water availability and the physical-chemical characteristics of the water.

### 3.2. Microbiological Analysis

Contrary to the observations of the physical-chemical parameters, the microbiological parameters were frequently above the recommended values, meaning that coliforms were detected in 100 mL of treated water up to 200 CFU/100mL (Figure 4). Coliforms were observed for all the household samples and most of the sampling dates, indicating that it is not a sporadic event. Again, no correlation was observed between the period of water availability and the coliforms’ abundance.

## 4. Discussion

All the physical-chemical parameters and heavy metals determined were within the values recommended, not evidencing any problem that could jeopardize public health (Figure 2). Only three household samples presented higher values of Cu (Punto 5, 6, and 7), although within the recommended values, which may be a consequence of the age or material of the pipes, something we could not validate. 

Regarding the microbiological parameters, the presence of coliforms was observed for all the sampled points and most of the sampling dates. These results show that fecal contamination is affecting most of the water distribution system of the city of Ilave. This same problem was previously described, mainly in rural areas in Peru [17]. Heitzninger et al. [17] described that in many Peruvian houses, practices such as the use of water storage containers, and the way the water is collected from there, influences the microbiological quality of the water. However, what our study shows is that water is contaminated from when it arrives at the consumers’ houses. The most probable reason for the presence of coliforms in the drinking water is the low level of chlorine detected, below the 0.5 mg/L recommendation of the health authorities which is considered to be potable [19,25]. The importance of maintaining a free chlorine level in treated drinking water to avoid the occurrence of coliforms is well described [26]. The local authorities should guarantee that the concentration of the disinfectant, along all the distribution system, is high enough to avoid microbial contamination, which is usually reached by doing re-chlorination along the system. Lee and Schwab [14] have already described the absence of re-chlorination steps, that assure the maintenance of a residual concentration of the disinfectant along all of the distribution system, as one of the major problems in developing countries. Fortunately, this is a problem that has already started to be solved. Previous descriptions for other Peruvian cities, such as Puno, show that the levels of residual chlorine is already adequate [27].

Previous studies reported that microbial contamination is frequent in intermittent water service systems [28,29,30,31,32]. Possible explanations for that are the low levels of residual disinfectant commonly detected in these systems [33,34,35] and the bacteria regrowth and biofilm formation due to a higher water stagnation [29]. Considering this, we hypothesized that houses with a shorter period of water service (4 h/day versus 8 h/day) could have a lower water quality. However, contrary to the hypothesis formulated initially, no significant correlation (*p* > 0.05) between water contamination and the period of water availability was observed. These results show that urgent actions must be taken by the city of Ilave to provide drinking water with the microbiological quality required to avoid possible occurrences of water-related and water-borne diseases. Access to drinking water of good quality (without microbiological contaminants) is essential for improving the health safety and quality of life of those populations.

To increase water quality monitoring, it is also important to have a better overview of the real situation. In this study, the samples were collected just in one season. It would be important to validate the results also in different periods of the year to understand if other factors, such as temperature or rain, may also affect the drinking water quality.

## 5. Conclusions

The water distributed to the city of Ilave, presented good physical-chemical, but not microbiological quality. The presence of coliforms in the water should be a direct consequence of the low levels of residual chlorine detected (below the recommended values). The differences in the period of water availability (4 h/day or 8 h/day) did not interfere with the drinking water quality, with there not being an observed correlation between the water availability and the microbiological quality of the water.

From this study, two major recommendations can be drawn. The importance of: (i) including additional steps of re-chlorination along the drinking water distribution system, and (ii) defining regular monitoring for the early detection of the occurrence of microbiological contamination in drinking water.

## Figures and Tables

**Figure 1 ijerph-19-10779-f001:**
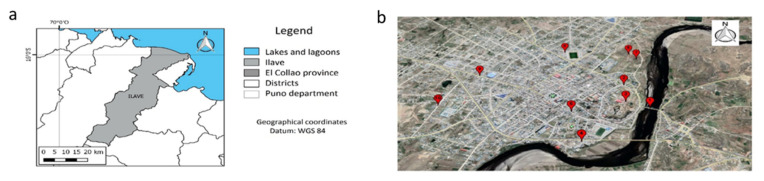
Location of the district of Ilave in the province of El Collao (**a**) and location of the ten sampling points (**b**), located in the city of Ilave, Region of Puno, Peru.

**Figure 2 ijerph-19-10779-f002:**
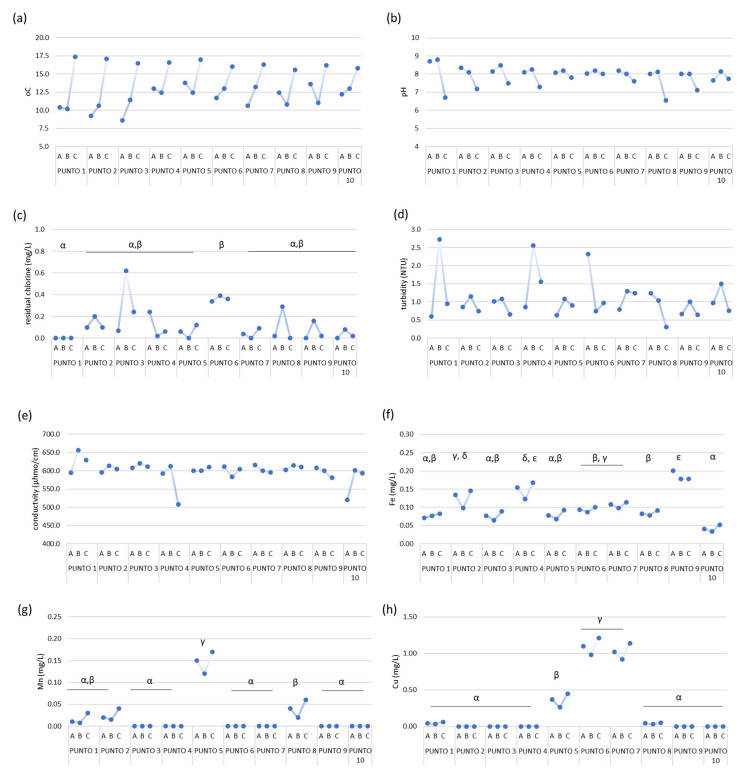
Determination of physical-chemical parameters for the 10 sampling points and three sampling dates: (**a**) temperature, (**b**) pH, (**c**) chlorine concentration, (**d**) turbidity, (**e**) conductivity, and the heavy metals concentration of (**f**) Fe, (**g**) Mn, and (**h**) Cu. Zn was not detected in the samples and Ni was only detected in samples A from Punto 2 and 3, at 0.001 and 0.003 mg/L, respectively. When observed, significant differences (*p* < 0.05) between samples are indicated by the letters α, β, γ, ε, and δ.

**Figure 3 ijerph-19-10779-f003:**
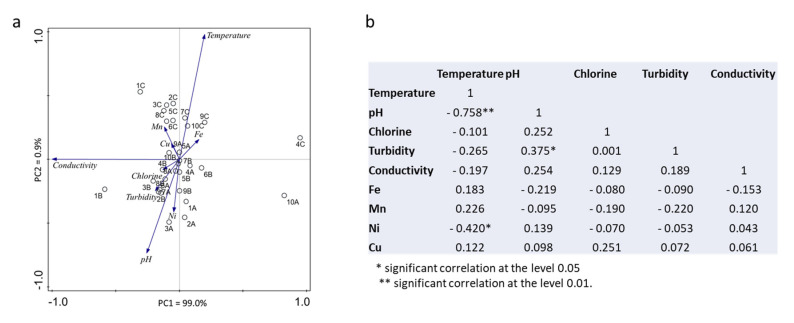
Principal Components analysis (PCA) (**a**) and Pearson correlation coefficients (**b**) determined for the physical-chemical parameters.

**Figure 4 ijerph-19-10779-f004:**
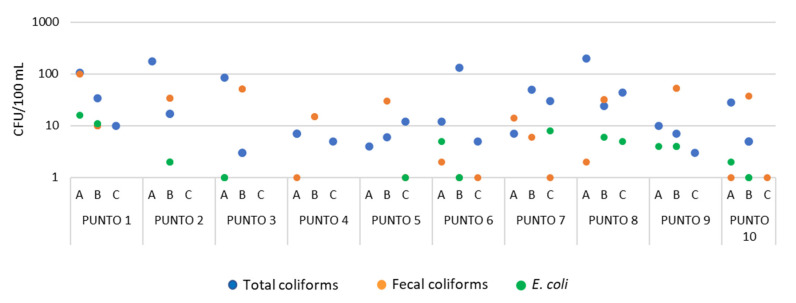
Abundance of coliforms in the ten sampling points.

**Table 1 ijerph-19-10779-t001:** Details of the 10 sampling points.

Sampling Point	Sampling Points Designation	Address: Street/Av.	Georeference UTM			Type of Sampling Point	Water Availability (h/Day)
			East	North	Altitude		
PUNTO 1	Rio	Old Bridge	432798	8221380	3645	River	n.a.
PUNTO 2	Primera Vivienda	St. Zepita	432572	8221401	3874	Household	4
PUNTO 3	Barrio Santa Barbara	St.16 de Agosto	432505	8221462	3866	Household	4
PUNTO 4	Barrio San Sebastian	St. San Sebastian	432311	8220792	3847	Household	4
PUNTO 5	Complejo	St. Desaguadero	432417	8222090	3843	Household	8
PUNTO 6	Plaza De Armas	St. 28 de Julio	432017	8222145	3868	Household	8
PUNTO 7	Barrio San Martin	St. Pachacutec	432556	8221979	3867	Household	8
PUNTO 8	Cusupi	Av. Republica	431933	8222027	3860	Household	8
PUNTO 9	Terminal Terrestre	Av. Puno	431109	8221753	3866	Household	8
PUNTO 10	Barrio Alto Alianza	Av. Primavera	431158	8221189	3857	Household	8

**Table 2 ijerph-19-10779-t002:** Legal limits allowed in water used for human consumption according to the Peruvian regulation (DS N° 031-2010-SA) [25].

Parameter	Legal Limit (DS N° 031-2010-SA, MINSA 2011)
Physical-chemical	
turbidity	5 NTU
residual chlorine	250 mg Cl^−^/L
temperature	n.a.
conductivity (25 °C)	1500 μhmo/cm
pH	6.5–8.5
Microbiological	
total coliforms	0 CFU/100 mL at 35 °C
fecal coliforms and *E. coli*	0 CFU/100 mL at 44.5 °C
Heavy metals	
Zn	3.0 mg/L
Mn	0.4 mg/L
Ni	0.02 mg/L
Fe	0.3 mg/L
Cu	2.0 mg/L

n.a., not applicable; NTU, nephelometric turbidity units; CFU, colony-forming units.

## Data Availability

All the raw data is available on request.

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
