# Peer review of "Quality Control of Drinking Water in the City of Ilave, Region of Puno, Peru"

_ijerph, 2022, doi:10.3390/ijerph191710779_

Round 1

Reviewer 1 Report

Dear Authors,

The study monitored the water quality from the region of Puno, trying to determine the correlation between physical-chemical and microbiological parameters. The manuscript is well written and brings relevant information about water quality for the local population. However, the study goal does not significantly improve the intended field of research.

Author Response

Dear Authors,

The study monitored the water quality from the region of Puno, trying to determine the correlation between physical-chemical and microbiological parameters. The manuscript is well written and brings relevant information about water quality for the local population. However, the study goal does not significantly improve the intended field of research.

Answer: Dear reviewer 1, thank you for the time dedicated to the review of the manuscript. We understand the concern with the lack of innovation, but we believe this study may have a significant relevance for the water quality management in Peru and other countries with a similar condition. Peru is a country with still a high health concern related with waterborne diseases and where the population still does not have a full access to drinking water. The understanding of the water quality condition and the parameters that may be improved to improve the populations health and quality of life are of major importance.

Reviewer 2 Report

<The terminology “Physical chemical” should be spelled/written consistently throughout the paper. 

Line 32-33: <For that reason, the access to clean water is one of the United Nations Sustainable Development Goals for 2030 (https://sdgs.un.org/goals).> This needs to be appropriately cited.  

Line 34-36: <Therefore, an adequate supply of drinking water to drink and wash food is critical to reducing the transmission of diseases (4)>. The statement is somewhat confusing. For instance, how does the supply quantity (for drinking use) help reduce diseases transmission? The authors are also advised to check the source material.   

Line 42-46: <In order to avoid waterborne diseases, governments promote the disinfection of drinking water through sanitary regulations; being one of the most common disinfectants the chlorine (11,12), mainly because it has a residual effect that allows a longer effect and a monitoring, which has been adopted and recommended by the World Health Organization (3,12).> This needs to be rewritten for clarity. Two or three statements can be useful.  

In 231-232, the authors reported as follows “the differences in the period of water availability did not interfere with the drinking water quality”. The authors need to elaborate this for clarity. 

I advise the authors to include a section outlining the structure of the paper and this may be added in the end of the introduction.

The authors reported as follows “the Coliforms that should be absent from drinking water were detected in all the 22 household samples” What are the potential factors affecting coliforms? It would be useful to complement this finding from scholarly literatures. Additionally, more specific water quality issues and factors affecting water quality from the study area needs to be reviewed as part of introduction.

Line 211-214: <However, what our study shows is that water is contaminated since if arrives the consumers house and the most probable reason for the presence of coliforms in drinking water is the low level of chlorine detected in the water, below the 0.5 mg/L recommended by the health authority to be considered as potable (24,30).> This needs to be rephrased for readers clearly. 

Line 222-224: <Contrary to the hypothesis formulated initially, these results show that urgent actions must be taken by the city of Ilave to provide drinking water with the microbiological quality required to avoid possible occurrences of water-related and water-borne diseases.> If this is the case, such hypothesis should be discussed. 

Line 225-226 <Access to the drinking water of good quality is essential for improving the health safety 225 and quality of life of those populations.> “Good quality” needs to be defined. 

The water sampling was carried out only in September, 2019.  It is likely that water quality may not be the same throughout the year. For instance, it may be affected during the rainy season, especially in river water/open sources. This poses a question on sampling frequency/method and the results may be misleading. How do you justify this? It would also be useful to present/discuss the sampling standard set by the Ministry of Health or related regulatory agencies? 

The authors drew water sample from 9 household taps and a river. In line 23-25, they claim the need for adding steps of re-chlorination along the distribution system to ensure the chlorine level. The authors need to clearly discuss the water supply technology (e.g., pumping, gravity etc.). In other words, how was the river water tapped and distributed to these households? To make this even simpler, what is the current distribution systems and measures taken to manage water quality?

The correlations between physical-chemical parameters were determined using the Pearson correlation coefficient. It would be useful to discuss briefly the rationale for choosing the method. 

I recommend the authors to: I) include a table showing correlation results, ii) elaborate the discussion with feedback from previous similar research, and iii) add research limitations as a separate section or as part of conclusion. 

This paper demands recommendations for water quality management and monitoring. Some of the key recommendations are to be drawn as part of conclusion. 

Most of the “references” contains error and this needs to be addressed.   

Author Response

<The terminology “Physical chemical” should be spelled/written consistently throughout the paper. 

Answer: Thank you for the correction. The text was revised accordingly.

Line 32-33: <For that reason, the access to clean water is one of the United Nations Sustainable Development Goals for 2030 (https://sdgs.un.org/goals).> This needs to be appropriately cited.  

Answer: The reference to the UN report was added to replace the website.

Line 34-36: <Therefore, an adequate supply of drinking water to drink and wash food is critical to reducing the transmission of diseases (4)>. The statement is somewhat confusing. For instance, how does the supply quantity (for drinking use) help reduce diseases transmission? The authors are also advised to check the source material.   

Answer: We agree with the reviewer that the sentence was not well written. We updated the bibliography and rephrased to: “Therefore, an adequate supply of drinking water with good quality to drink and wash food is critical to reducing the transmission of diseases”.

Line 42-46: <In order to avoid waterborne diseases, governments promote the disinfection of drinking water through sanitary regulations; being one of the most common disinfectants the chlorine (11,12), mainly because it has a residual effect that allows a longer effect and a monitoring, which has been adopted and recommended by the World Health Organization (3,12).> This needs to be rewritten for clarity. Two or three statements can be useful.  

Answer: Thank you for the suggestion. The sentence was fragmented in three sentences, and we hope the text is clearer now: “To avoid waterborne diseases, governments promote the disinfection of drinking water through sanitary regulations. The use of chlorine as a disinfectant is among the most commonly used treatments (12,13). One of the big advantages of chlorine as a disinfectant is the residual effect that allows a longer effect and a monitoring, which has been adopted and recommended by the World Health Organization (4,13).”.

In 231-232, the authors reported as follows “the differences in the period of water availability did not interfere with the drinking water quality”. The authors need to elaborate this for clarity. 

Answer: The sentence was reformulated and now reads “The differences in the period of water availability (4 h/day or 8 h/day) did not interfere with the drinking water quality, being not observed a correlation between the water availability and the microbiological quality of the water.”

I advise the authors to include a section outlining the structure of the paper and this may be added in the end of the introduction.

Answer: Maybe we are not well understanding the reviewer suggestion, but we think the journal guidelines do not allow that we include a kind of “table of contents”. Also, considering that the paper is short, maybe that is not needed. However, if the reviewer believes that it will significantly improve the paper readability, we can include it.

The authors reported as follows “the Coliforms that should be absent from drinking water were detected in all the 22 household samples” What are the potential factors affecting coliforms? It would be useful to complement this finding from scholarly literatures. Additionally, more specific water quality issues and factors affecting water quality from the study area needs to be reviewed as part of introduction.

Answer: As we refer in the Discussion of the results, we believe that the low levels of chlorine is the major reason for the presence of coliforms in the water. The same was also reported, for developing countries, in other studies “Lee and Schwab (20) have already described the absence of re-chlorination steps, that assure the maintenance of a residual concentration of the disinfectant along all the distribution system, as one of the major problems in developing countries.”. Extra bibliography was added to the text to support this.

In the Introduction we describe the major factors affecting water quality in Peru according to the bibliography available “In 2018, just 52% of the Peruvian households had access to safe drinking water, being observed inequalities in the access according to the size of the city and the economic condition of the house owners (18). In Peru, it is still very common that the water service is intermittent with interruptions in the distribution. The stoppage of the water pump is a way to reduce the expense of electrical power (19). However, it may result in the degradation of the quality of the water supplied to the users (20). In peri-urban communities, fecal contamination of drinking water is still very common in Peru (21–25).”.

Line 211-214: <However, what our study shows is that water is contaminated since if arrives the consumers house and the most probable reason for the presence of coliforms in drinking water is the low level of chlorine detected in the water, below the 0.5 mg/L recommended by the health authority to be considered as potable (24,30).> This needs to be rephrased for readers clearly. 

Answer: There was a typo in the sentence. We rephrased and now reads “However, what our study shows is that water is contaminated since it arrives at the consumers' houses. The most probable reason for the presence of coliforms in drinking water is the low level of chlorine detected in the water, below the 0.5 mg/L recommended by the health authorities to be considered potable (25,31).”.

Line 222-224: <Contrary to the hypothesis formulated initially, these results show that urgent actions must be taken by the city of Ilave to provide drinking water with the microbiological quality required to avoid possible occurrences of water-related and water-borne diseases.> If this is the case, such hypothesis should be discussed. 

Answer: The text was improved and now the formulated hypothesis is better described “Previous studies reported that microbial contamination is frequent in intermittent water service systems (34–38). Possible explanations for that are the low levels of residual disinfectant commonly detected in these systems (39–41) and the bacteria regrowth and biofilm formation due to a higher water stagnation (35,42). Considering this, we hypothesized that houses with a shorter period of water service (4 h/day versus 8 h/day) could have lower water quality. However, contrary to the hypothesis formulated initially, no significant correlation (p>0.05) between water contamination and period of water availability was observed.”

Line 225-226 <Access to the drinking water of good quality is essential for improving the health safety 225 and quality of life of those populations.> “Good quality” needs to be defined. 

Answer: The sentence now reads “The access to drinking water of good quality (without microbiological contaminants), is essential for improving the health safety and quality of life of those populations.”

The water sampling was carried out only in September, 2019.  It is likely that water quality may not be the same throughout the year. For instance, it may be affected during the rainy season, especially in river water/open sources. This poses a question on sampling frequency/method and the results may be misleading. How do you justify this? It would also be useful to present/discuss the sampling standard set by the Ministry of Health or related regulatory agencies? 

Answer: We agree with the reviewer that to monitor the water quality in other seasons would strengthen the results. That is now discussed in the paper “It is also important to highlight the importance to increase water quality monitoring to have a better overview of the real situation. In this study, the samples were collected just in one season. It would be important to validate the results also in different periods of the year to understand if other factors, such as temperature or rain, may also affect the drinking water quality.”

The authors drew water sample from 9 household taps and a river. In line 23-25, they claim the need for adding steps of re-chlorination along the distribution system to ensure the chlorine level. The authors need to clearly discuss the water supply technology (e.g., pumping, gravity etc.). In other words, how was the river water tapped and distributed to these households? To make this even simpler, what is the current distribution systems and measures taken to manage water quality?

Answer: The information about the distribution system is described in the methods section “The drinking water system (DWS) is conventional, with water being distributed by gravity and treated by flocculation, decantation, filtration and disinfection (with chlorination system through a drip dispenser or constant flow with pump).”. However, we don’t have any additional information for example about the distance from the chlorination points and the sampled taps.

The correlations between physical-chemical parameters were determined using the Pearson correlation coefficient. It would be useful to discuss briefly the rationale for choosing the method. 

I recommend the authors to: I) include a table showing correlation results, ii) elaborate the discussion with feedback from previous similar research, and iii) add research limitations as a separate section or as part of conclusion. 

Answer: The Pearson correlation coefficient allows to determine linear correlation between two sets of data, that was what we tried to do here. We rephrased the text that now reads “The existence of a linear correlation between physical-chemical parameters was determined using the Pearson correlation coefficient.”.

A matrix with the Pearson correlation coefficients was added to Figure 3. Always possible the discussion was enriched with results from previous similar research and the limitations of the study, namely the ones related with just one season have been sampled, was added to the Discussion.

This paper demands recommendations for water quality management and monitoring. Some of the key recommendations are to be drawn as part of conclusion. 

Answer: As suggested we included the key recommendation in the Conclusions section, that now reads “From this study, two major recommendations can be drawn. The importance of: i) include additional steps of re-chlorination along the drinking water distribution system, and ii) define regular monitoring to early detect the occurrence of microbiological contamination in drinking water.”

Most of the “references” contains error and this needs to be addressed.   

Answer: The list of references was carefully revised and all the errors detected corrected.

Reviewer 3 Report

1) How were sample sites selected? Just one sample was acquired at each sample site each day?

2) At household sampling points, where was the sample collected? Was it taken where the water comes into the house or from some other source, such as a kitchen/bathroom tap? 

3) Why did a multiparameter bench photometer was used for heavy metal analysis instead of atomic absorption spectrophotometry? Have you noticed that the Ni, Fe, and Cu detection limits were higher than those limits accepted by the regulation?

4) Figure 2, PUNTO 10 in all plots is missing. 

5) Figure 2b, Could you explain the pH variation at PUNTO 1? What happens in the river?

6) Figure 2f, In Figure 2f, the values plotted of Mn, Fe, and Cu are below the equipment detection limits, Mn (0.3 mg/L), Fe (0.4 mg/L), and Cu (5 mg/L); therefore, with what accuracy do they measure and quantify these elements concentration below the equipment detection limit when by itself the detection limit specifies than the equipment is not able to determine below that concentration. It is necessary to review this data since they conclude that no elements, such as Ni are present in more than one sample. Still, the detection limit for this element is higher (1 mg/L) than the maximum permitted by the legislation (0.2 mg/L). 

7) Please check Figure 4. How is it possible to count more fecal coliforms than total coliforms? or more E. coli than fecal coliforms?

8) Once all the data has been thoroughly checked and amended, further work should be done on discussion and conclusions. 

Author Response

1) How were sample sites selected? Just one sample was acquired at each sample site each day?

Answer: The selection of the sample sites was done trying to have a good geographic distribution. From each sampling location was just collected one sample, on three different days with a 7-day interval between each sample, to try to normalize possible variances along time. Now this was made clearer in the text “The water sampling was carried out during October 2019, on three consecutive weeks (11, 18, and 25 October), and three measurements of each parameter evaluated.”

2) At household sampling points, where was the sample collected? Was it taken where the water comes into the house or from some other source, such as a kitchen/bathroom tap? 

Answer: The samples were collected directly from the kitchen tap. This information was added to the text that now reads “The household water samples were collected from the kitchen tap, to have a water sample closer to the water that is consumed.”

3) Why did a multiparameter bench photometer was used for heavy metal analysis instead of atomic absorption spectrophotometry? Have you noticed that the Ni, Fe, and Cu detection limits were higher than those limits accepted by the regulation?

Answer: Unfortunately, an atomic absorption spectrophotometer was not available. We apologize for the misunderstanding, but the limits referred in the text are the maximum limits not the minimum limits of quantification, as can be validated in the manual of the instrument, page 127 (http://www.keison.co.uk/products/hannainstruments/manHI_83200.pdf). The text was revised to avoid this misunderstanding.

4) Figure 2, PUNTO 10 in all plots is missing.

Answer: Figure 2 was corrected.

5) Figure 2b, Could you explain the pH variation at PUNTO 1? What happens in the river?

Answer: We believe that what happen at Punto 1 (the river) is the same that happens at the other points, an increase in the temperature that leads to a reduction in the pH.

6) Figure 2f, In Figure 2f, the values plotted of Mn, Fe, and Cu are below the equipment detection limits, Mn (0.3 mg/L), Fe (0.4 mg/L), and Cu (5 mg/L); therefore, with what accuracy do they measure and quantify these elements concentration below the equipment detection limit when by itself the detection limit specifies than the equipment is not able to determine below that concentration. It is necessary to review this data since they conclude that no elements, such as Ni are present in more than one sample. Still, the detection limit for this element is higher (1 mg/L) than the maximum permitted by the legislation (0.2 mg/L). 

Answer: As answered before the indicated limits of the instrument are maximum limits. Once again, we apologize for this misunderstanding.

7) Please check Figure 4. How is it possible to count more fecal coliforms than total coliforms? or more E. coli than fecal coliforms?

Answer: The counts of total coliforms, fecal coliforms and E. coli were, for most of the samples, in the same range, with a maximum variance of 1log-unit, which is not a significant variance in terms of bacteria count, that are always variable. Also, the use of different selective culture media for each of the target bacterial groups introduces some variability. However, we agree that it may represent some lack of accuracy of the methodology used and for that reason we discussed the results always in terms of total coliforms. For water quality evaluation, the regulations require the absence of any of those bacterial groups in 100mL of sample, meaning that the detection of any of them represents a health issue.

8) Once all the data has been thoroughly checked and amended, further work should be done on discussion and conclusions. 

Answer: All the manuscript was revised and improved considering the comments of the four reviewers. We hope now it is significantly improved.

Reviewer 4 Report

In the manuscript ID ijerph-1786592, Ferro et al. reported data on the drinking water quality of a town in Peru. The importance of these data is mainly associated to the sanitary risk for the inhabitants.

The topic is interesting and fits the aims of IJERPH and of the special issue. However, I think the manuscript needs major revisions before being acceptable for publication. Since the dataset is relatively small and results were poorly analysed and discussed, this article could become a valuable short communication.

My major comment is related to statistical analyses that should be corrected and clearly shown in the text. I think ANOVA or Kruskal-Wallis, followed by Tukey test or Mann-Whitney test respectively, can be used to detect differences of each parameter considered among the ten sampling sites. This should be specified in the methods and the statistical results should be reported and discussed. Moreover, I did not understand how you tested the correlation among the water quality and the period of water availability. I think you can test the difference among points 2-3-4 (4 h/day) and points 5-6-7-8-9-10 (8 h/day) using a non parametric test such as Mann-Whitney.

I suggest also improving the quality of figures, mainly of figure 2 (avoid repetition of the names of parameters (title or y-axis), and summarize the figure caption).

The discussion should be extended for instance including more comparisons with water quality in other countries of the world, with areas facing similar issues. Are all the water quality criteria adequate and similar to those adopted in other countries?  

E. coli in italics. Check English grammar mainly in the discussion section.

Author Response

In the manuscript ID ijerph-1786592, Ferro et al. reported data on the drinking water quality of a town in Peru. The importance of these data is mainly associated to the sanitary risk for the inhabitants.

The topic is interesting and fits the aims of IJERPH and of the special issue. However, I think the manuscript needs major revisions before being acceptable for publication. Since the dataset is relatively small and results were poorly analysed and discussed, this article could become a valuable short communication.

Answer: We acknowledge the reviewer the time dedicated to review of the manuscript and suggest improvements. If the editorial board is of the same opinion, we also agree that the manuscript can be a short communication.

Round 2

Reviewer 2 Report

Thanks to the authors for their efforts in addressing my comments. At this stage I do not have any further feedback. 

Reviewer 4 Report

The authors improved the manuscript accounting for my comments. I think it can be accepted for publication in its current form. Considering the small dataset and the length of the manuscript, I suggest publishing it as short communication.